# Exploiting Adam-like Optimization Algorithms to Improve the Performance of Convolutional Neural Networks

**Loris Nanni**[1]                                            LORIS.NANNI@UNIPD.IT

**Gianluca Maguolo**[1]                          GIANLUCA.MAGUOLO@PHD.UNIPD.IT

**Alessandra Lumini**[2]                              ALESSANDRA.LUMINI@UNIBO.IT

[1] *DEI, University of Padua, viale Gradenigo 6, Padua, Italy*

[2] *DISI, Università di Bologna, Via dell'università 50, 47521 Cesena, Italy*

**Editors:** Under Review for MIDL 2021

## Abstract

We compare Adam based variants that rely on the difference between the present and the past gradients, where the step size is adjusted for each parameter. Besides, we propose new Adam based optimizers. We run several tests benchmarking the proposed methods using medical image data. The proposed ensemble obtains a performance comparable than the current state of the art. The MATLAB source code is available at https://github.com/LorisNanni/Exploiting-Adam-like-Optimization-Algorithms-to-Improve-the-Performance-of-Convolutional-Neural-Netw.

**Keywords:** Adam-like Optimization, Deep Networks Training, Ensemble.

## 1. Introduction

The minimization of the training loss of CNNs relies on algorithms based on gradient descent, the most naïve of which is SGD. Since SGD has some limitations, a lot of modifications have been proposed. In our experiments we compare different optimization approaches in CNNs for image classification tasks. Selecting a fixed architecture and training the net by varying the optimization process, we obtain a large number of networks that we use to create an ensemble. At first, we introduce some Adam-based variants for deep network optimization and we propose our new optimizers. The Adam based variants outperforms Adam, moreover, we show that changing the optimization is a feasible procedure for yields a set of different networks, providing a large number of partially independent classifiers.

## 2. Methods

We propose and evaluate three variants of Adam optimization method. DGrad is a variant of diffGrad (Dubey et al., 2019). It is based on the moving average of the element-wise squares of the parameter gradients. Cos#1 and Cos#2 are minor variant of DGrad based on the application of a cyclic learning rate (Smith, 2017) to DGrad. The proposed approaches have different methods for defining $\xi_t$, then the parameters are updated using Equation (20) defined in (Dubey et al., 2019). DGrad takes up the ideas of diffGrad defining the following absolute difference between two consecutive steps of the gradient:

$$\Delta ag_t = |g_t - avg_t| \tag{1}$$

where $avg_t$ contains the moving average of the element-wise squares of the parameter gradients. We then normalize $\Delta ag_t$ by its maximum and we define $\xi_t$ as:

$$\Delta \hat{ag}_t = \frac{\Delta ag_t}{\max(\Delta g_t)}, \qquad \xi_t = \sigma\left(4 \cdot \Delta \hat{ag}_t\right) \tag{2}$$

where $\sigma$ is the sigmoid function. The final update for each parameter $\theta_t$ of the network is as in equation (20) in (Dubey et al., 2019). The rationale of the "$4\times$" is to increase the range of the output of the sigmoid function.

Cos#1 and Cos#2 exploit the idea of using a cyclic learning rate. In Cos#1 we use the cosine function to define a range of variation of the learning rate and $lr_i$ is used as a multiplier of $\Delta \hat{ag}_t$ in the definition of $\xi_t$, which becomes:

$$lr_t = 2 - \left|\cos\left(\frac{\pi t}{steps}\right)\right| e^{-0.01(\mathrm{mod}(t, steps)+1)}, \qquad \xi_t = \sigma\left(4 \cdot lr_t \cdot \Delta \hat{ag}_t\right) \tag{3}$$

In Cos#2 the definition of $\xi_t$ is the following:

$$lra_t = \left(\left|\cos\frac{\pi t}{steps}\right| e^{-0.01(\mathrm{mod}(t, steps)+1)}\right) \tag{4}$$

$$\xi_t = \sigma\left(2 \cdot lr_t \cdot \Delta \hat{ag}_t\right) + \sigma\left(4 \cdot lra_t\right) - 0.5 \tag{5}$$

In this method, if $lr_t = 0$, we reset $lr_t = 0.009$ for that iteration. In both methods, the final update for the network parameters of the network $\theta_t$ is performed as in the Equation (20) defined in (Dubey et al., 2019).

All the CNNs have been trained using the following parameters: batch size $= 30$; Number of epochs $= 20$; global learning rate $= 0.001$; gradient decay factor $= 0.9$; squared gradient decay factor $= 0.999$. Data augmentation is performed considering random reflection and random scale on both axes.

## 3. Experiments and Discussion

The following datasets have been used: HeLa, the 2D HELA dataset (Boland and Murphy, 2001), with a 5-fold cross validation testing protocol; BG, the Breast Grading Carcinoma dataset (Dimitropoulos et al., 2017), with 5-fold cross validation; LAR, the Laryngeal dataset (Moccia et al., 2017) is already divided in 3 subfolders to be used for cross-validation. In Table 1, we report the results of stand-alone and ensemble models combined by sum rule. The numbers between parenthesis are the number of models in the ensemble. For ensemble models, we report the result of the sum rule, while for stand-alone models we report the average accuracy on 7 experiments. The performance reported in Table 1 shows that Adam methods obtain worse generalization than SGD. On average, the tested variants of Adam obtain better performance than original Adam. We used ResNet50 as backbone network.

From Table 1 we can see that the performance of Adam based approaches strongly improves considering ensemble of CNNs and that Adam variants, on average, perform better the original Adam. Besides, the fusion of CNNs trained with different optimization methods (i.e. SGD with Adam) allows to improve the performance: SGD(7) + DGrad(7) outperforms SGD(14) (ensembles with the same size). In our opinion, to combine networks trained using different optimization methods is a feasible way for building an ensemble.

Table 1: Performance of ensemble of CNN

| Accuracy | HeLA | BG | LAR |
|---|---|---|---|
| SGD | 92.09 | 88.33 | 93.03 |
| Adam | 55.90 | 86.57 | 92.15 |
| diffGrad | 79.00 | 89.00 | 93.01 |
| DGrad | 75.25 | 89.29 | 91.07 |
| Cos#1 | 78.92 | 88.38 | 92.19 |
| Cos#2 | 66.25 | 88.05 | 93.04 |
| Adam(7) | 74.30 | 89.67 | 96.29 |
| diffGrad(7) | 94.88 | 91.67 | 95.91 |
| DGrad(7) | 95.35 | 92.67 | 94.85 |
| Cos#1(7) | 95.00 | 92.67 | 95.38 |
| Cos#2(7) | 91.05 | 92.00 | 95.98 |
| DGrad(7) + Cos#1(7) | 95.81 | 92.33 | 95.91 |
| DGrad(7) + Cos#1(7) + diffGrad(7) | 96.28 | 92.33 | 96.06 |
| DGrad(14) | 95.70 | 92.67 | 95.68 |
| SGD(14) | 96.05 | 94.00 | 94.70 |
| SGD (7) | 95.70 | 94.00 | 94.32 |
| SGD(7) + DGrad(7) | 96.16 | 94.00 | 95.38 |
| SGD(14) + DGrad(7) + Cos#1(7) + diffGrad(7) | 96.98 | 94.33 | 96.14 |

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
