# OpenReview forum: "Exploiting Adam-like Optimization Algorithms to Improve the Performance of Convolutional Neural Networks"
_MIDL.io/2021/Conference/Short — MIDL 2021 Poster_

### Official Review · Reviewer_646B · 2021-04-23

**Confidence:** 4
**Final Rating:** 3

**Summary:**

The authors present an study testing well known optimizers, inclunding making an essemble composed by the same architecture trained by different optimizers, in three medical imaging datasets: HELA; BG and LAR. Additionally, the authors proposed variations to those well known optimizers named DGrad, Cos#1 and Cos#2. The authors conclude that an ensemble of the same method trained with different optimizers, and even mixing optimizers, is a feasible approach to making CNN ensembles.

**Strengths:**

The authors were able to summarize the experiments very well, while also detailing the modifications performed on DGrad, Cos#1 and Cos#2.

The used data is public and provided.

It is known that ensembles can improve performance especially when the performance of the single model is poor. The authors paid attention to include experiments of ensembles using the traditional optimizers and the proposed optimizers, showcasing the improvement might not be only because of using an ensemble, but also because of the use of the different optimization and the mix of optimizers.



**Weaknesses:**

I did not find a description of the used CNN architecture. Even though it did not vary between experiments, in my opnion it would be important to have at least a citation if its an off the shelf architecture, or a brief description.

The link for the open code is not going directly to the correct repository, instead its linking to a github user. You might want to change it.

The writing could benefit from improvements (see detailed comments).





**Deanonymize Review:**

yes

**Detailed Comments:**

Some minor writing mistakes:

In the introduction, 'create an ensemble'  instead of 'create ensemble'.

The last introduction phrase is confusing and could be rewritten.

A ':' is missing when you present equation (2).

End of Section 2: '[...] random scale on both axes' instead of 'both the axis'.



**Justification Of The Rating:**

The paper needs some minor writing adjustments for a strong acceptance, including some description of the used CNN architecture.

In my opnion this is a interesting experiment idea that would be of interest for the community as a poster presentation.



**Paper Type:**

both

**Special Issue:**

no

---

### Meta-Review · Area_Chair_5tyF · 2021-05-09

**Recommendation:** Accept (Poster)
**Confidence:** 5

**Metareview:**

The AC has carefully checked the paper and agrees with the reviewer that the paper contains ideas worth presenting at MIDL. However, the AC also strongly urges the authors to carefully check the syntax and grammar before submitting the camera-ready version.

---

### Decision · Program_Chairs · 2021-05-11

Accept (Poster)